# Improved Simulated-Daylight Photodynamic Therapy and Possible Mechanism of Ag-Modified TiO_2_ on Melanoma

**DOI:** 10.3390/ijms24087061

**Published:** 2023-04-11

**Authors:** Jing Xin, Jing Wang, Yuanping Yao, Sijia Wang, Zhenxi Zhang, Cuiping Yao

**Affiliations:** Key Laboratory of Biomedical Information Engineering of Ministry of Education, School of Life Science and Technology, Institute of Biomedical Analytical Technology and Instrumentation, Xi’an Jiaotong University, Xi’an 710048, China

**Keywords:** melanoma, simulated-daylight photodynamic therapy, Ag-doped TiO_2_, Ag-core TiO_2_

## Abstract

Simulated-daylight photodynamic therapy (SD-PDT) may be an efficacious strategy for treating melanoma because it can overcome the severe stinging pain, erythema, and edema experienced during conventional PDT. However, the poor daylight response of existing common photosensitizers leads to unsatisfactory anti-tumor therapeutic effects and limits the development of daylight PDT. Hence, in this study, we utilized Ag nanoparticles to adjust the daylight response of TiO_2_, acquire efficient photochemical activity, and then enhance the anti-tumor therapeutic effect of SD-PDT on melanoma. The synthesized Ag-doped TiO_2_ showed an optimal enhanced effect compared to Ag-core TiO_2_. Doping Ag into TiO_2_ produced a new shallow acceptor impurity level in the energy band structure, which expanded optical absorption in the range of 400–800 nm, and finally improved the photodamage effect of TiO_2_ under SD irradiation. Plasmonic near-field distributions were enhanced due to the high refractive index of TiO_2_ at the Ag-TiO_2_ interface, and then the amount of light captured by TiO_2_ was increased to induce the enhanced SD-PDT effect of Ag-core TiO_2_. Hence, Ag could effectively improve the photochemical activity and SD-PDT effect of TiO_2_ through the change in the energy band structure. Generally, Ag-doped TiO_2_ is a promising photosensitizer agent for treating melanoma via SD-PDT.

## 1. Introduction

Melanoma is a commonly occurring severe skin malignancy induced by melanocytes [1]. The incidence of melanoma is ever-increasing. It is traditionally considered to be metastatically invasive as it can invade and spread to neighboring tissues [2]. Additionally, it is resistant to chemotherapeutic drugs and radiation therapy [3,4]. Therefore, more effective therapeutic strategies for melanoma need to be developed [5,6]. Photodynamic therapy (PDT) can selectively destroy diseased cells or tissues as they are more sensitive to light irradiation [7]. During PDT, a photosensitizer in a singlet ground state undergoes visible or near-infrared irradiation, absorbs energy, and attains an excited triplet state through intersystem crossing [8]. The triplet state of the photosensitizer reacts with oxygen or the substrate through electron/hydrogen atom or energy transfer processes, producing reactive oxygen species, especially singlet oxygen, to damage biological components (e.g., amino acids, unsaturated lipids, and DNA bases) [9]. Because singlet oxygen can diffuse by only 10–20 nm during its lifetime of 0.01–0.04 µs, the intracellular damage targets of PDT are very close to the intracellular localization of the photosensitizer. Therefore, PDT might be a non-invasive, effective treatment strategy for melanoma cancer therapy [10].

PDT efficiency relies on three primary factors: the photosensitizer, light, and molecular oxygen, as per the PDT mechanism [11,12]. The frequently utilized light sources for PDT include coherent light sources (argon and argon-pumped lasers, solid-state lasers, metal vapor-pumped dye lasers, and optical parametric oscillator lasers) as well as non-coherent light sources (fluorescent lamps, halogen lamps, metal halide lamps, xenon arc lamps, and phosphor-coated sodium lamps) [13]. Specifically, coherent light sources can provide high-power output and are widely used in PDT [14]. However, they can cause severe pain, erythema, and edema during irradiation. These side effects are usually intolerable and can even make patients refuse treatment [15,16]. To solve this problem, daylight PDT or simulated-daylight PDT has attracted much attention. Although it cannot penetrate deep tissues, it is very safe, nearly painless, well-tolerated, and mostly nonsurgical [17,18,19,20,21]. Because skin disease is a superficial disease, it has no limitation of tissue depth. Using daylight PDT for skin disease is more effective than using traditional laser PDT [22]. Some studies have shown that although daylight PDT is effective for some skin diseases, it has no obvious therapeutic effect on melanoma [22,23]. This might be because the response of existing photosensitizers to daylight is very low in melanoma. Therefore, specific photosensitizers with a high response to daylight need to be developed to treat melanoma.

Titanium dioxide (TiO_2_) is a typical biocompatible semiconductor oxide metallic nanomaterial, which is used worldwide for different applications [24,25,26,27,28]. In 1985, Wake et al. used TiO_2_ to kill microbes through photochemical sterilization under metal halide lamp irradiation, which indicated that TiO_2_ could be used in the field of PDT [29,30,31,32]. TiO_2_ is photoactive in the presence of UV light, which provides a basis for its application in daylight PDT [24,33,34]. However, TiO_2_ is ineffective as a daylight photodynamic therapeutic agent for treating cancer as it responds poorly to sunlight. To shift the TiO_2_ absorption spectrum to the visible region to expand the daylight response range, several approaches have been proposed [35,36]. Among them, doping with the metal ions using transition metal or non-metal ions to change the optoelectronic features of TiO_2_ could significantly shift the optical response of TiO_2_. In addition, modifying with plasmonic metallic nanoparticles to combine the photocatalytic properties of TiO_2_ and the optical properties of plasmonic nanoparticles could extend the photocatalytic activity of TiO_2_ from UV light to visible or even to the NIR range of radiation. Among all metallic materials, silver (Ag) exhibits the most interesting physical properties and unique optical properties [37]. Hence, in this study, Ag-modified TiO_2_ nanomaterials with varying structures (Ag-doped TiO_2_ and Ag-core TiO_2_) were synthesized to improve the limited SD-PDT effect on melanoma by increasing the daylight response. The improvement in the photochemical activity and the therapeutic effect of PDT were compared and the possible mechanisms were theoretically studied. Generally, the described TiO_2_ modification method based on Ag significantly increased the photochemical properties of the metallic nanomaterials. The synthesized Ag-doped TiO_2_ was found to be a promising agent for treating melanoma using daylight PDT.

## 2. Results

### 2.1. Synthesis and Characterization of Ag-Modified TiO_2_ with Different Structure

To increase the light response range of TiO_2_ and improve the simulated-daylight PDT effect of TiO_2_ on melanoma, Ag-modified TiO_2_ was synthesized. Based on the different optical properties of nanomaterials with different structures, Ag-doped TiO_2_ and Ag-core TiO_2_ were synthesized, respectively. The TEM results of the synthesized TiO_2_ were comparable to P25 (the commercialized TiO_2_ nanoparticles) purchased from Sigma, and Ag-doped TiO_2_ showed that the size of the sphere was 100 nm (Figure 1A,B). Nonetheless, the absorption spectrum of the Ag-doped TiO_2_ displayed a prominent redshift (Figure 2A), which may be attributed to alterations in the energy band structure. The XRD findings demonstrated that the lattice structure of TiO_2_ remained unaltered following Ag being doped into TiO_2_ (Figure 2B). In Ag-core TiO_2_, the addition of sodium bicarbonate to the reaction mixture led to the formation of a TiO_2_ shell enveloping Ag nanoparticles. The crystal lattice structure of TiO_2_ with a spacing of 0.32 nm appeared after calcination (Figure 1C,D). The TEM results revealed that the synthesized silver had a uniform particle size (Figure 1E). The thickness of the shell of TiO_2_ increased with the increase in the concentration of sodium bicarbonate to 1.5 mL. When the concentration of sodium bicarbonate was 0.9 mL and 1.3 mL, the thickness of the TiO_2_ shell was about 5 nm and 18.7 nm, respectively (Figure 1F,G). When the concentration of sodium bicarbonate was 1.5 mL, the shell of TiO_2_ agglomerated (Figure 1H). The results of the energy spectrum analysis from TEM-EDS also confirmed that silver was successfully coated by TiO_2_ (Ag 81.43%, Ti 7.36%, and O 11.21%) (Figure 2D). The results of the absorption spectrum showed that the Ag-doped TiO_2_ showed absorption in the range of 400–800 nm, which was similar to the absorption of the synthesized TiO_2_ and higher than the absorption of P25 (Figure 2A). The Ag-core TiO_2_ showed a prominent red shift and a decrease in the intensity of the absorption spectrum with an increase in the thickness of the TiO_2_ shell (Figure 2C).

### 2.2. Comparative Analysis of the Photocatalytic Activity of Ag-Modified TiO_2_ with Different Structures

The photochemical activities of Ag-modified TiO_2_ were evaluated through the photocatalytic degradation of methylene blue. As shown in Figure 3A, methylene blue could be degraded under TiO_2_ induction after Ag was added to Ag-doped TiO_2_, but this degradation effect was not observed for free TiO_2_. Upon reaching a 2% Ag concentration, the degradation rate rose from 40% to 70%. As shown in Figure 3B, methylene blue was degraded by 62.3% and 36.3% under the induction of Ag-core TiO_2_ (5 nm thick) and P25, respectively. The catalytic efficiency of Ag-core TiO_2_ decreased as the thickness of the shell increased, and the Ag-core TiO_2_ (5 nm thick) of the shell had the highest catalytic efficiency. Hence, Ag-modified TiO_2_ significantly increased the photochemical activity of TiO_2_. Compared to Ag-core TiO_2_, Ag-doped TiO_2_ exhibited better photochemical activity.

### 2.3. Comparative Analysis of the Cytotoxicity Assay and Phototoxicity of Ag-Modified TiO_2_ with Different Structures

To evaluate the safety and photodamage effects of the synthesized Ag-doped TiO_2_ and Ag-core TiO_2_, we determined the viability of A375 cells (human melanoma cell line) without irradiation and with irradiation, by performing a CCK-8 assay. Cell viability induced by 50 µg/mL TiO_2_ in different reagents was higher than 90%, and no noticeable difference was found between them, which indicated that the synthesized reagents had negligible dark cytotoxicity in A375 cells. When the concentration of TiO_2_ decreased, the cell viability increased further (Figure 4A). Hence, the synthesized Ag-doped TiO_2_ and Ag-core TiO_2_ were safe for usage. As shown in Figure 4B, after irradiation by daylight, 1 µg/mL TiO_2_ in the different reagents could not inhibit A375 cells, and 50 µg/mL TiO_2_ in the different reagents could strongly inhibit them. Specifically, the simulated-daylight photodamage effect of Ag-doped TiO_2_ was higher than that of Ag-core TiO_2_. For example, cell viability decreased to 24.5% and 31.6% after induction by Ag-doped TiO_2_ and Ag-core TiO_2_, respectively. However, cell viability only decreased to 67.6% and 60.1% after induction by P25 and the synthesized free TiO_2_ at 50 µg/mL. 

The photodamage effect increased with the increase in the irradiation dosage (Figure 3C). However, after irradiation with 40 J/cm^2^, the untreated cells could also be partially inhibited, and the cell survival rate of A375 was only about 71%. Based on the principle of little toxic effect on normal tissue occurring to the greatest extent during PDT, the irradiation dosage for the synthesized Ag-modified TiO_2_ could be controlled below 40 J/cm^2^. Overall, Ag could effectively improve the photodamage effect of TiO_2_ under simulated-daylight irradiation, and Ag-doped TiO_2_ had a more significant daylight PDT effect on melanoma.

### 2.4. Comparative Analysis of ROS Generation by Ag-Modified TiO_2_ with Different Structures

During PDT, generated ROS can lead to the damage of cellular components and then induce cell death. Hence, the ability to generate ROS determines the PDT effect. To determine the simulated-daylight PDT effect induced by Ag-modified TiO_2_, the ability to generate ROS was evaluated using a DCFH-DA probe—the most widely used probe for detecting intracellular H_2_O_2_ and oxidative stress. As per the fluorescence imaging findings, ROS were generated after induction by P25, the synthesized TiO_2_, and Ag-modified TiO_2_, and the induction was higher when Ag-modified TiO_2_ was used (Figure 5A). The fluorescence intensity of DCFH-DA increased obviously in Ag-modified TiO_2_ (Figure 5B). Compared with P25, Ag-core TiO_2_ and Ag-doped TiO_2_ resulted in significant increases, at average values of 1.75-fold and 1.95-fold, respectively. Hence, the ROS levels induced by Ag-doped TiO_2_ were higher than those induced by Ag-core TiO_2_. The inhibitory activity induced by all reagents containing TiO_2_ could be effectively weakened by using a quenching agent of ROS. After being treated with P25, the synthesized TiO_2_, Ag-core TiO_2_, and Ag-doped TiO_2_, as well as 20 mM histidine, cell activities increased from 67.6%, 60%, 31.6%, and 24.5% to 85.3%, 80.8%, 60.2%, and 55.6%, respectively. This revealed that the degree of inhibition induced by Ag-modified TiO_2_ was higher than that induced by the synthesized TiO_2_ or P25 (Figure 5C). Hence, our findings showed that Ag-modified TiO_2_, especially Ag-doped TiO_2_, efficiently improved simulated-daylight PDT by enhancing the ability to generate ROS.

### 2.5. The Theoretical Mechanistic Analysis

The results of the experiment showed that modification with Ag could effectively enhance the photodamage effect of TiO_2_ under simulated-daylight irradiation. Specifically, Ag-doped TiO_2_ had a more significant daylight PDT effect on melanoma. This might be due to an increase in the response of TiO_2_ to daylight facilitated by Ag. To confirm whether this mechanism was used, changes in the optical absorption properties of TiO_2_ induced by doping with Ag were used, based on density functional theory using the CASTEP code. First, a supercell of TiO_2_ (containing 16 Ti atoms and 32 O atoms) and 2% Ag-doped TiO_2_ (containing 15 Ti atoms, 32 O atoms and 1 Ag atom) was constructed and used as the later calculation model (Figure 6A,B). Then, the changes in the band structure and the density of the states of pure TiO_2_ and Ag-doped TiO_2_ were calculated. The energy of the band gap of pure TiO_2_ was 3.325 eV, which then decreased to 3.14 eV when Ag was doped (Figure 6C,D). Additionally, compared to the band structure of pure TiO_2_, two new impurity energy levels were introduced, and one of them passed through the Fermi level, which indicated that it can act as an acceptor at the shallow impurity energy level as a bound state of a hole (Figure 6C,D). This shallow acceptor impurity level could enhance the separation of electron-hole pairs, produce free conduction holes, decrease the recombination of photo-generated electrons and holes, and decrease the energy required for the electrons to escape. The computed results of the total density of states and partial density of states of pure TiO_2_ and Ag-doped TiO_2_ showed that the 4d-orbital electrons of Ag atoms were used to introduce the new impurity energy levels and produce the new electronic states through strong mixing with the p-orbital states of oxygen atoms (Figure 7). Hence, these two impurity energy levels of Ag-doped TiO_2_ mainly occurred due to doping with Ag. Generally, the doped Ag atom has an influence on the TiO_2_ energy structure and induced bandgap narrowing. Ag is the main reason for the the increase in the daylight response of Ag-doped TiO_2_. Ag-doped TiO_2_ expanded optical absorption in the range of 400–800 nm and improved the photodamage effect of TiO_2_ under daylight irradiation. 

First principles analysis can be used to analyze the properties of materials with a crystal structure but not of materials with a core–shell structure. To determine the cause of the improvement in the SD-PDT effect of TiO_2_ induced by Ag-core TiO_2_, discrete dipole approximation simulations were studied because these changes might be induced by the localized surface plasmon resonance enhancement effect. First, a series of Ag-core TiO_2_ complexes with a constant Ag core and TiO_2_ shells of different thicknesses (0 nm, 5 nm, 10 nm, 15 nm, and 20 nm) were calculated. As shown in Figure 8A, the results showed that the absorption spectrum has an obvious red shift and a significant decrease in the intensity with an increase in the thickness of the TiO_2_ shell. These results matched the absorption spectrum data we measured. Plasmonic near-field distribution showed a strong change in the Ag-core TiO_2_ (Figure 8B). The electric near-field intensities were considerably enhanced due to the high refractive index of TiO_2_ at the Ag- TiO_2_ interface. With the increase in the electric near-field intensity, the amount of light captured by TiO_2_ and the PDT effect increased. The relationship between the field enhancement effect and the thickness of the TiO_2_ shell results showed that the field enhancement effect gradually weakened and the ability of the TiO_2_ shell to capture light decreased with an increase in the thickness of the shell, which occurred probably because the shell affected the movement of photo-generated electrons and holes (Figure 8C). Hence, the Ag-core TiO_2_ with a 5 nm thick shell had the highest photocatalytic efficiency.

## 3. Discussion

Photodynamic therapy is an effective therapeutic strategy for skin disease in clinical therapy [38]. However, it is often accompanied by severe adverse effects during treatment. A large number of patients are unable to continue treatment due to these adverse effects. In 2008, daylight PDT was first introduced as a less painful, outdoors alternative to conventional PDT, with similar clinical effectiveness [39]. However, daylight PDT efficacy was often dependent on weather conditions. For example, in the U.K., daylight PDT is practical between the months of March or April and September or October, when the temperature is above 10 °C in the day (from 9:00 to 18:00) and the fluence rate reaches 130 W/m^2^ [40]. In addition, to avoid patient exposure to harmful wavelengths of ultraviolet radiation during daylight PDT, organic sunscreens should be used to prevent sun damage. To provide a controlled, daylight PDT environmental setting and remove the disadvantage of exposure to harmful ultraviolet radiation, SD-PDT has been investigated using an indoor daylight-simulating lamp. Wulf and co-workers reported that four different lamp candidates (18 W red-, 140 W red-, and 50 W white-light-emitting diode lamps and halogen lamps from 250 W slide projectors as well as 400 W overhead projectors for SD-PDT were able to photobleach a PPIX photosensitizer completely [39]. Calzavara-Pinto et.al. revealed that SD-PDT using a lamp with an output confined to the red waveband (630 ± 5 nm) and a polychromatic white LED lamp (400–700 nm) can represent a valid therapeutic method for Actinic cheilitis [41]. In our study, the SD-PDT effect can be obtained under the irradiation of a sunlight Xenon lamp with an emission spectral range of 380 nm to 700 nm. Hence, a sunlight Xenon lamp (380–700 nm) is also a useful lamp candidate for SD-PDT. However, in our study, we did not evaluate and compare the SD-PDT effect of Ag-modified TiO_2_ under other lamp sources. In further studies, more detailed comparative research may be needed to obtain a better SD-PDT anti-tumor therapeutic effect.

In this study, an investigation to improve the strategy to increase the simulating-daylight response of existing photosensitizers is the main purpose of the research that we want. TiO_2_ is a potent oxygen radical generator. However, it is limited in SD-PDT by the necessity to use ultraviolet irradiation with low tissue penetration and its harmful impact on the human body. To maximize the visible light absorption of TiO_2_, inorganic compounds were usually doped to the TiO_2_ during their preparation, because this process can narrow the bandgap in the TiO_2_ nanoparticle’s structure and decrease the necessary activation energy. Among these inorganic compounds, noble metals (such as gold (Au), silver (Ag), platinum (Pt), and palladium (Pd)) were used to dope TiO_2_, one after another [33]. All absorption ranges of TiO_2_ were shifted to longer wavelengths and enhanced photocatalytic activities under visible light were obtained to different degrees after doping. However, compared with the other noble metals used, Ag has been regarded as a better candidate due to its higher catalytic activity and ROS generation ability [41,42]. Hence, Ag-doped TiO_2_ may be suitable for daylight PDT or SD-PDT. Unfortunately, there are few study reports that shows that Ag-doped TiO_2_ is used in daylight PDT or SD-PDT. However, Alshamsan et.al. revealed that Ag-doped TiO_2_ has the potential to selectively kill cancer cells while sparing normal cells through ROS generation in HepG2 (human liver cancer cell line) [43]. It gave us a reason to conduct research and evaluate the SD-PDT effect of Ag-doped TiO_2_ on melanoma. In this study, the results showed that the limited photochemical activity and SD-PDT effect of TiO_2_ could be improved significantly through doping Ag to the TiO_2_. In addition, our results showed that the degree of the improvement photochemical activity was independent of the concentration introduction of Ag into TiO_2_. This may be caused by the synthesized Ag-doped TiO_2_ complex having different light responses with different concentrations of Ag under simulated-daylight irradiation. This change in light response was not entirely dependent on the doped Ag concentration. Lu and co-workers measured Ag-doped TiO_2_ photocatalysts with different concentrations of Ag (1–5%) in their previous study. They reported that 2% Ag-doped TiO_2_ had the highest photocatalytic activity under ultraviolet radiation and 5% Ag-doped TiO_2_ had the highest photocatalytic activity under solar light [40]. Hence, the introduction of the different concentrations of Ag into TiO_2_ may cause the different changes in light response. Generally, in our study, Ag-doped TiO_2_ with a certain concentration of Ag efficiently improved TiO_2_ photochemical activity compared with TiO_2_.

Besides the Ag-doped TiO_2_ complex, TiO_2_-coated Ag nanoparticles have found applications in many fields because they can combine the surface plasmon resonance properties of a Ag core and the photoactivity of the TiO_2_ shell [44]. Tunable optical properties can be obtained through a change in the ratio of the core radius and shell thickness. Hence, Ag-core TiO_2_ was also usually used to increase the optical absorption of TiO2 and extend its absorption region to that of visible light. As with Ag-doped TiO_2_, there are few study reports that show that Ag-core TiO_2_ is used in daylight PDT or SD-PDT, and the improvement in the photocatalysts’ effect was not compared directly between Ag-doped TiO_2_ and Ag-core TiO_2_ in any other study. To find a better improvement strategy to increase the simulating-daylight response of a TiO_2_ photosensitizer, the improvement in the photocatalytic activity and SD-PDT effect induced by Ag-doped TiO_2_ and Ag-coreTiO_2_ was compared. The results showed that the described TiO_2_ modification method based on Ag significantly increased the photochemical properties of TiO_2_ In addition, the synthesized Ag-doped TiO_2_ was found to be a promising agent for treating melanoma using daylight PDT, and doping Ag to TiO_2_ is the optimal enhanced strategy.

Several studies revealed that the introduction of Ag into TiO_2_ improves TiO_2_’s photochemical activity due to two mechanisms. (1) Ag can act as an electron acceptor to increase the separation efficiency of a photogenerated electron-hole pair because its Fermi level is below the conduction band of TiO_2_; (2) The generation of a local surface plasmon resonance effect extends the visible light absorption range and increases the photocatalytic efficiency of TiO_2_ [45]. Hence, in this study, first principles analysis was performed for Ag-doped TiO_2_ and the discrete dipole approximation for the Ag-core TiO_2_ was calculated. The results showed that a new shallow acceptor impurity level appeared in the energy band structure of Ag-doped TiO_2_, which decreased the recombination of photo-generated electrons and holes and the energy needed for the excitation of electrons. This expanded the light response range of TiO_2_ and made it more responsive to sunlight. A strong field enhancement effect was obtained at the interface between the TiO_2_ shell and the Ag core of Ag-core TiO_2_, which increased the amount of light captured by TiO_2_ and improved its photochemical activity. These are consistent with the previously described mechanism. These further confirm the reliability of our research on this improvement strategy to increase the simulating-daylight response of a TiO_2_ photosensitizer. Hence, Ag-doped TiO_2_ is a promising photosensitizer agent for treating melanoma with daylight PDT.

## 4. Material and Methods

### 4.1. Reagents and the Cell Lines

Silver nitrate (AgNO_3_), tetra butyl titanate, and P25 (TiO_2_ nanoparticles) were purchased from Sigma. TiCl_3_ was purchased from Aladdin. Absolute ethanol, sodium citrate, solidum borohydride (NaBH_4_), butyl alcohol, sodium bicarbonate (NaHCO_3_), and N-butanol were purchased from Tianjin Fuyu Chemical Co., Ltd. and Tianjin Tianli Chemical Reagent Co., Ltd. (Tianjin, China). A Cell Counting Kit (CCK-8) was purchased from Dojindo (Japan). DCFH-DA was purchased from Beyotime Company (Shanghai, China). Human melanoma cell line A375 was obtained from the Cell Bank of the Chinese Academy of Sciences (Shanghai, China) [46]. The A375 cells were cultured in DMEM medium (HyClone) supplemented with 10% fetal bovine serum (HyClone) and 1% penicillin/streptomycin in a humidified incubator at 37 °C with 5% CO_2_.

### 4.2. Synthesis and Modification of TiO_2_


Ag-doped TiO_2_: Nanosized TiO_2_ was synthesized using the sol-gel process. Briefly, 5 mL of glacial acetic acid and 20 mL of absolute ethanol were added to 3 mL of ddH_2_O and stirred at room temperature for 30 min. A total of 5 mL of tetra-butyl titanate and 10 mL of absolute ethanol were mixed and added to the above solution. After stirring for 30 min, the solution was dispersed ultrasonically for 20 min, left to stand for 24 h at room temperature, and then dried at 80 °C for 12 h. The obtained dried product was fully milled and then cauterized to 450 °C in a muffle for 2 h. Purified TiO_2_ was finally obtained. As with TiO_2_, 3 mL of 25 mg, 50 mg, 75 mg, or 100 mg AgNO_3_ and 5 mL of glacial acetic acid were added in turn to 20 mL of absolute ethanol and stirred at room temperature for 30 min and then added to 15 mL of a tetra-butyl titanate solution containing 10 mL of absolute ethanol. After stirring, dispersing, drying, milling, and cauterizing, the Ag-doped TiO_2_ with a mixing ratio of 1%, 2%, 3%, and 4% was obtained. 

Ag-core TiO_2_: The Ag nanoparticles were produced utilizing the seed growth method. Initially, a 20 mL 1% sodium citrate solution diluted with 75 mL of ddH_2_O was stirred for 15 min at 70 °C. Next, 1.7 mL of 1% AgNO_3_ and 2 mL of 0.1% NaBH_4_ were added and stirred for 1 h at the same temperature. This process led to the creation of a 4 nm sized silver seed solution. Second, 2 mL of sodium citrate solution diluted with 80 mL of ddH_2_O was stirred for 15 min at 130 °C. Next, 10 mL of silver seed solution and 1.7 mL of AgNO_3_ were added in turn, stirred for 1 h at the same temperature, and then centrifuged repeatedly. The purified Ag nanoparticles were yielded. Next, 0.3 mL of TiCl_3_, the different concentrations of 0.2 M sodium bicarbonate (0.9 mL, 1.1 mL, 1.3 mL, and 1.5 mL) and the silver nanoparticle solution were added in turn to 8 mL of ddH_2_O, stirred for 30 min, and then washed with ddH_2_O and absolute ethanol, respectively. Furthermore, 10 mL of N-butanol was added. The mixing solution was heated in oil baths at 100 °C for 10 min and then dried, milled, and cauterized. Finally, Ag-core TiO_2_ with TiO_2_ shells of different thicknesses were synthesized. 

### 4.3. Characterization of Ag-Doped TiO_2_ and Ag-Core TiO_2_

The morphologies of Ag-doped TiO_2_ and Ag-core TiO_2_ were observed using transmission electron microscopy (TEM; JEM-2100, JEOL, Tokyo, Japan). Absorption spectra of Ag-doped TiO_2_ and Ag-core TiO_2_ were recorded using an ultraviolet–visible spectrophotometer (V-550 UV/VIS, JASCO, Tokyo, Japan). X-ray diffraction was conducted using an X-ray diffractometer (XPert Powder, PANalytical B.V. Netherlands). An energy dispersive spectrometer was used to observe the distribution pattern of various elements (Ag, Ti, and O) using TEM-EDS (JEM-2100 Plus, JEOL Ltd., Japan), operating with an accelerating voltage of 200 kV. The photocatalytic capability of Ag-doped TiO_2_ (1%, 2%, 3%, and 4%) and Ag-core TiO_2_ (5 nm, 10 nm, 15 nm, and 20 nm) were evaluated through the photocatalytic degradation of methylene blue under a simulated sunlight Xenon lamp with an emission spectral range of 380 nm to 700 nm. The irradiation time was 10 min at 665 nm. The absorption intensity was recorded using an ultraviolet–visible spectrophotometer (V-550 UV/VIS, JASCO, Tokyo, Japan).

### 4.4. Cell Viability Analysis

The cytotoxicity assay and phototoxicity assay to evaluate the safety and simulated-daylight PDT effect on melanoma cells (A375 cell line) were measured using a CCK-8 assay (Cell Counting Kit-8—allows for sensitive colorimetric assays for the determination of cell viability in cell proliferation and cytotoxicity assays). Briefly, A375 cells (8000/well) were seeded to sterile 96-well flat-bottomed plates and incubated overnight in a humidified incubator at 37 °C with 5% CO_2_. Diluted Ag-doped TiO_2_ and Ag-core TiO_2_ with the different concentrations (1 μg/mL, 5 μg/mL, 10 μg/mL, 20 μg/mL, 50 μg/mL) were added to corresponding cells in the 96-well flat-bottomed plates. After incubation for 6 h, the medium containing reagent was replaced by fresh cell culture medium. For the cytotoxicity experiment, the plates were then incubated for 24 h in a humidified incubator. In the phototoxicity experiment, the cells were irradiated with a sunlight Xenon lamp at 30 J/cm^2^ for 15 min and then incubated for 12 h in a humidified incubator. To assess the influence of the irradiation dose on the phototoxicity effect, the cells were treated with 50 μg/mL of the Ag-doped TiO_2_ or Ag-core TiO_2_ previously mentioned. Then, they were irradiated with the daylight Xenon lamp at 10 J/cm^2^, 20 J/cm^2^, 30 J/cm^2^, 40 J/cm^2^, or 50 J/cm^2^ for 15 min before being incubated for 12 h in a humidified incubator. Finally, all the treated cells were measured for absorbance levels at 450 nm using a microplate reader (Infinite M200 Pro., Tecan, Switzerland). The absorbance levels of cells were calculated as
$$\frac{\mathrm{OD}\;\mathrm{of}\;\mathrm{treated}\;\mathrm{group}-\mathrm{OD}\;\mathrm{of}\;\mathrm{blank}\;\mathrm{control}\;\mathrm{group}}{\mathrm{OD}\;\mathrm{of}\;\mathrm{control}\;\mathrm{group}-\mathrm{OD}\;\mathrm{of}\;\mathrm{blank}\;\mathrm{control}\;\mathrm{group}}\times 100\%$$
OD is optical density.

### 4.5. Generation of ROS 

ROS is an indirect factor to induce cell damage on PDT. Therefore, the ability of the generation of ROS was measured with a DCFH-DA probe using a Nikon eclipse Ti fluorescence microscope (Nikon, Japan). Briefly, 2.5 × 10^3^ A375 cells were seeded to sterile 3.5 mL flat-bottomed plates and incubated overnight in a humidified incubator at 37 °C with 5% CO_2_. Then, the cells were treated with diluted Ag-doped TiO_2_, Ag-core TiO_2_, the synthesized TiO_2,_ and P25 for 6 h, washed with PBS twice, irradiated with the sunlight Xenon lamp at 40 J/cm^2^, incubated with 10 μmol/L DCFH-DA for 20 min at 37 °C in complete darkness, washed with PBS again, and then imaged using a FACScan system or detected using a fluorescence spectrophotometer under the excitation of 488 nm light. To quantify the ability of the generation of ROS, the fluorescence intensity of DCFH-DA was measured using a fluorescence spectrophotometer to detect the concentration of ROS in cells after being treated by Ag-doped TiO_2_, Ag-core TiO_2_, the synthesized TiO_2,_ and P25 under a simulated-sunlight Xenon lamp irradiation. The treated cells under the same conditions as above were harvested, incubated with 10 μmol/L DCFH-DA for 10 min at 37 °C in complete darkness, and then centrifuged, washed with PBS, and measured using a fluorescence spectrophotometer under an excitation of 488 nm light. In order to reveal the role of ROS more directly in simulated-daylight PDT induced by Ag-modified TiO_2_, inhibition tests were measured using a quenching agent of ROS (histidine). After being treated with the different agents containing TiO_2_, the cells were treated with 20 mM histidine for 30 min, washed with PBS, and then irradiated with the sunlight Xenon lamp. The cell activity was detected using CCK-8 analysis, as before.

### 4.6. First-Principles Analysis for Ag-Doped TiO_2_

Based on crystallographic principles, the shape, electronic environment, and other parameters of the crystal cell will change when some atoms are substituted with allochthonous atoms in this cell. Therefore, the change in sunlight response induced by Ag-doped TiO_2_ is most likely because some Ti atoms are substituted with Ag atoms. The energy band structure, the total density of states and the partial density of states, and the optical absorption properties were determined via density functional theory using the CASTEP code [47]. 

### 4.7. Discrete Dipole Approximation for Ag-Core TiO_2_

It has been reported that the photocatalysis of TiO_2_ could be enhanced using metal particle doping, polymer nanocomposites, core–shell nanoparticles, and so on, based on the localized surface plasmon resonance enhancement effect [48]. These electric field enhancement factors can be quantified and analyzed using simulations based on discrete dipole approximation [49]. Hence, theoretical mechanistic analysis of the change in sunlight response induced by silver-core TiO_2_ in this study was employed using discrete dipole approximation simulations using the DDSCAT program.

## 5. Conclusions

To improve the limited effect of daylight PDT on melanoma due to the low daylight response of commonly used photosensitizers, Ag-mediated TiO_2_ nanomaterials with different structures were synthesized, and then the improvement in the photocatalytic activity and PDT effect of these nanomaterials were compared. As per the findings, Ag effectively and significantly increased the photochemical activity and the PDT effect of TiO_2_ under simulated-daylight irradiation. Ag-doped TiO_2_ exhibited superior photocatalytic activity and a greater daylight-PDT-induced anti-tumor effect compared to Ag-core TiO_2_. To determine the mechanism, first principles analysis was conducted utilizing Ag-doped TiO_2_, whereas the discrete dipole approximation for Ag-core TiO_2_ was calculated. The results showed that doping Ag into TiO_2_ led to the formation of a new shallow acceptor impurity level in the energy band structure, which then enhanced the separation of electron-hole pairs, produced free conduction holes, reduced the recombination of photo-generated electrons and holes, and decreased the energy required for electrons to escape. This increased optical absorption in the range of 400–800 nm, which improved the photodamage effect of TiO_2_ under simulated-daylight irradiation. The plasmonic near-field distribution increased due to the high refractive index of TiO_2_ at the Ag- TiO_2_ interface, which increased the amount of light captured by TiO_2_ and enhanced the induction of the daylight PDT effect. In addition, when the thickness of the shell increased, the shell affected the movement of photo-generated electrons and holes, which decreased the overall photochemical activity. Overall, Ag proved to be highly effective in enhancing the photochemical activity and PDT effect of TiO_2_ when exposed to simulated-daylight irradiation on melanoma. Thus, Ag-doped TiO_2_ exhibits great potential as a photosensitizer agent for treating melanoma with daylight PDT.

## Figures and Tables

**Figure 1 ijms-24-07061-f001:**
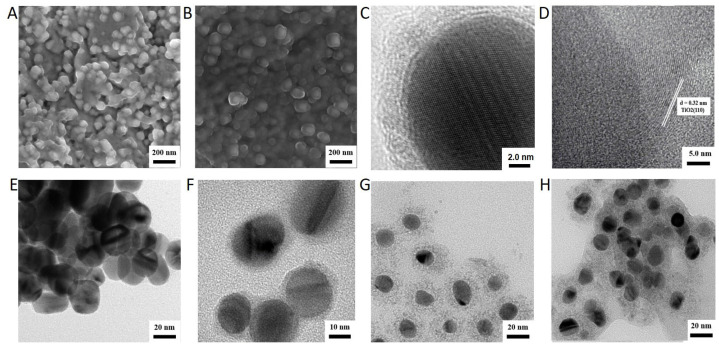
Transmission electron microscopy image of Ag-modified TiO_2_. (**A**) Transmission electron microscopy image of TiO_2_; (**B**) Transmission electron microscopy image of Ag-doped TiO_2_; (**C**) Transmission electron microscopy image of Ag-core TiO_2_ with a high resolution before the calcination treatment; (**D**) The crystal lattice structure of TiO_2_ in Ag-core TiO_2_ observed using transmission electron microscopy image with a high resolution after the calcination treatment; (**E**) Transmission electron microscopy image of Ag; (**F**) Transmission electron microscopy image of Ag-core TiO_2_ with 5 nm thick TiO_2_ shell (sodium bicarbonate at 0.9 mL); (**G**) Transmission electron microscopy image of Ag-core TiO_2_ with 20 nm thick TiO_2_ shell (sodium bicarbonate at 1.3 mL); (**H**) Transmission electron microscopy image of Ag-core TiO_2_ with shell of TiO_2_ agglomerated (sodium bicarbonate at 1.5 mL).

**Figure 2 ijms-24-07061-f002:**
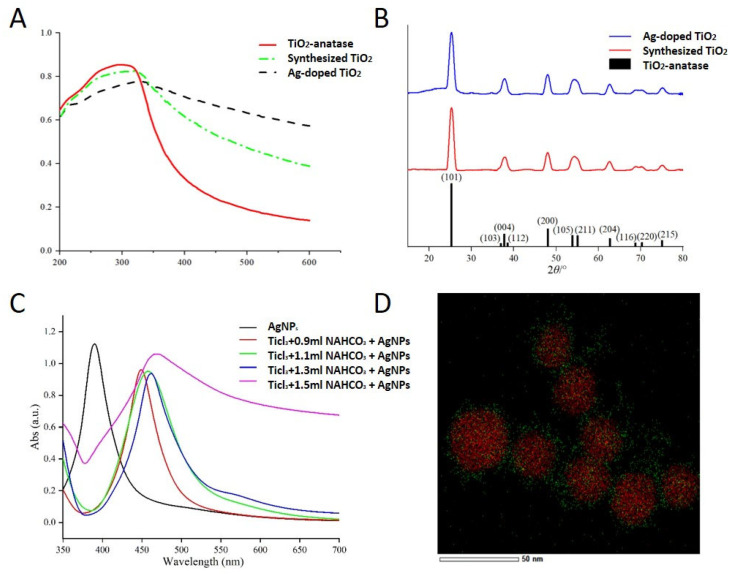
Properties of Ag-modified TiO_2_. (**A**) UV-vis absorption spectra of Ag-doped TiO_2_ compared with commercially available TiO_2_ (P25) and the synthesized TiO_2_; (**B**) XRD of Ag-doped TiO_2_ compared with P25 and the synthesized TiO_2_; (**C**) UV-vis absorption spectra of Ag-core TiO_2_ with the different thickness of the shell; (**D**) TME-EDS of Ag-core TiO_2_.

**Figure 3 ijms-24-07061-f003:**
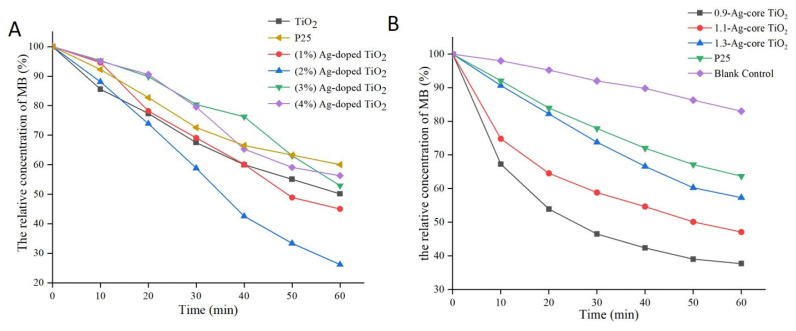
The photocatalytic degradation of methylene blue (MB) induced by Ag-doped TiO_2_ (**A**) and Ag-core TiO_2_ (**B**).

**Figure 4 ijms-24-07061-f004:**
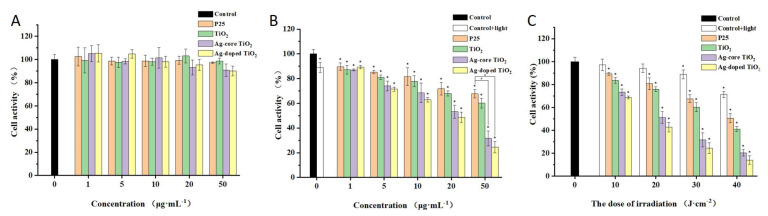
The cytotoxicity assay and phototoxicity of Ag-modified TiO_2_ through a CCK-8 assay; (**A**) Cell viability without irradiation—the cytotoxicity assay of Ag-modified TiO_2_ compared with P25 and the synthesized TiO_2_; (**B**) Cell viability after irradiation by daylight—the phototoxicity assay of Ag-modified TiO_2_ compared with P25 and the synthesized TiO_2_ with different concentrations of TiO_2_; (**C**) Cell viability after different irradiation dosage—the phototoxicity assay of Ag-modified TiO_2_ compared with P25 and the synthesized TiO_2_. *, *p* < 0.05, represents statistically significant difference between P25, the synthesized TiO_2_, Ag-core TiO_2_, the Ag-doped TiO_2_ group, and the control group.

**Figure 5 ijms-24-07061-f005:**
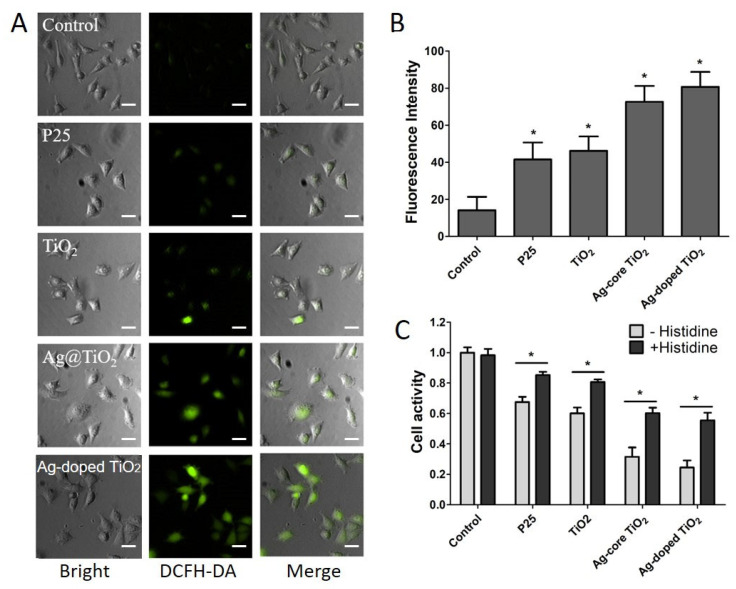
The ROS generation induced by Ag-core TiO_2_ and Ag-doped TiO_2_ and ROS inhibition using histidine compared with P25 and the synthesized TiO_2_; (**A**) Fluorescence imaging of ROS generation using DCFH-DA probe on A375 cells; (**B**) Fluorescence intensity assay of ROS generation for DCFH-DA probe on A375 cells. (**C**): ROS inhibition using 20 mM histidine for 30 min on A375 cells. The bar is 20 μm or SD. *, *p* < 0.05, represents statistically significant difference between the treated histidine group and the untreated histidine group in P25, the synthesized TiO_2_, Ag-core TiO_2_, Ag-doped TiO_2_, and control.

**Figure 6 ijms-24-07061-f006:**
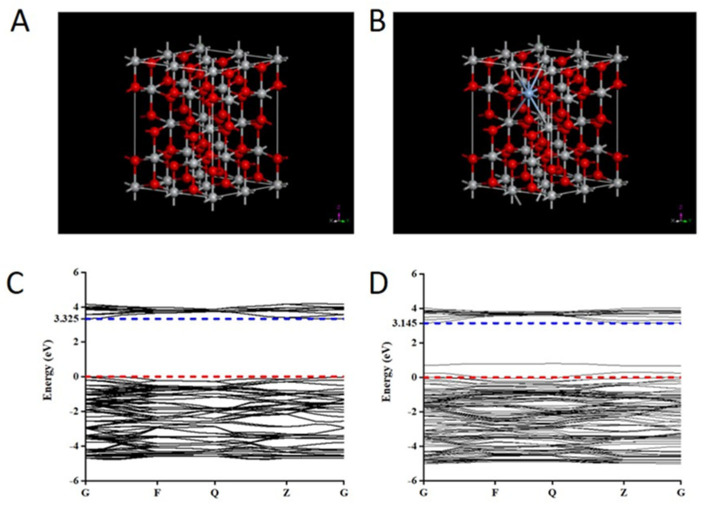
The crystalline structures of TiO_2_ and Ag-doped TiO_2_ (**A**,**B**) and the band structure of TiO_2_ and Ag-doped TiO_2_ (**C**,**D**) obtained through density functional theory analysis. The abscissa axis is in the indicated Brillouin zone for tetragonal structure of TiO_2_. The dashed red lines (energy zero) represent the valence-band maximum. The blue lines represent the minimum band gap at the G point.

**Figure 7 ijms-24-07061-f007:**
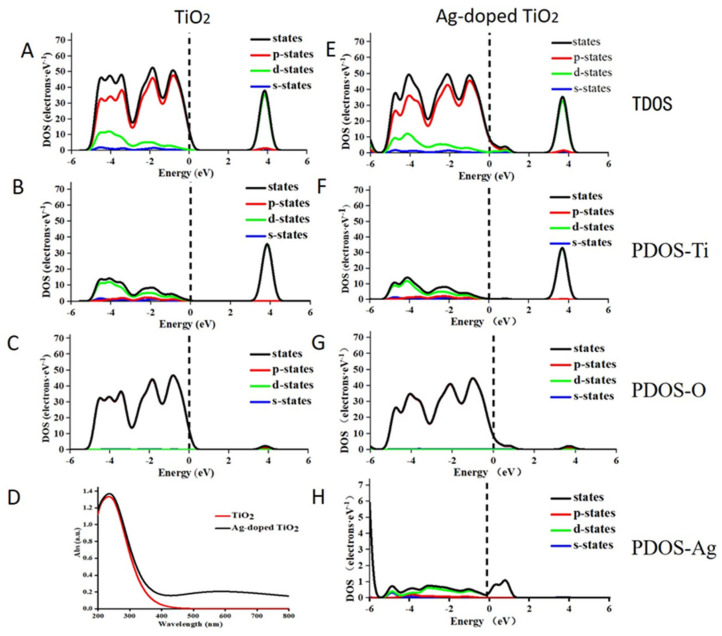
The total density of states (DOS) and the partial density of states (PDOS) for the projected states of Ti, O, and Ag corresponding to TiO_2_ (**A**–**C**) and Ag-doped TiO_2_ (**E**–**H**), and the calculated optical absorption spectrum of TiO_2_ and Ag-TiO_2_ (**D**) obtained through density functional theory analysis.

**Figure 8 ijms-24-07061-f008:**
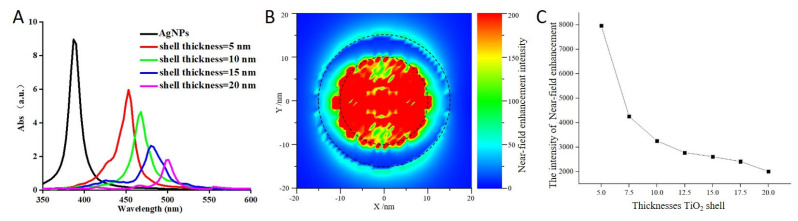
The simulated absorption spectrum (**A**) and the near-field enhancement distribution (**B**) and enhancement intensity (**C**) of Ag-core TiO_2_ with the increasing thickness of the TiO_2_ shell (from 5–20 nm).

## Data Availability

Data sharing not applicable.

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
