# Peer review of "Improved Simulated-Daylight Photodynamic Therapy and Possible Mechanism of Ag-Modified TiO2 on Melanoma"

_ijms, 2023, doi:10.3390/ijms24087061_

Round 1

Reviewer 1 Report

The article from Cuiping Yao et al. entitled “Improved simulated-daylight-photodynamic therapy and possible mechanism of Ag-modified TiO2 on Melanoma” deals with the preparation of Ag-doped TiO2 and Ag-core TiO2 nanoparticles for Stimulated-daylight-photodynamic therapy (SD-PDT). The authors prepared these nanoparticles and characterised them by TEM, DRX and UV-Vis. They assessed the photodegradation of the nanoparticles using methylene blue and evaluated their phototoxicity and ROS generation under sunlight irradiation in cell lines. Then, the authors related the photophysical and structural properties by density functional studies. The aims of the work are clear and well-presented using several techniques/methods. Overall, the study deserves of publication in IJMS although some points shall be solved beforehand:

(1) The authors shall revise the English. There are some typos along the text, i.e., “sliver” line 340.

(2) The theoretical mechanistic section must be explained better since it is difficult to follow their analysis and conclusions.

(3) The ROS generation study is not well-suited since the authors have to quantify the fluorescence emission somehow. Just the observation of fluorescence in a well-plate full of cells is not significant. The authors shall re-plan and re-do the assay.

(4) The methodology is not explained. It shall be completed, in this regards cell viability assay is not possible to understand. There is some part of the text missed.

(5) Why do the authors used A375 cells?

(6) Some parts of the text are duplicated, for instance lines 176-180 and lines 42-46. The authors shall be rewritten these parts.

Author Response

Revised manuscript submitted to International Journal of Molecular Science

Manuscript ID: IJMS-2299785

Title: Improved simulated-daylight-photodynamic therapy and possible mechanism of Ag-modified TiO2 on Melanoma

Authors: Jing Xin, Jing Wang, Yuanping Yao, Sijia Wang, Zhenxi Zhang and Cuiping Yao*

Dear Editors and Reviewers,

  Thank you very much for your evaluation and comments from the reviewers for our manuscript. We have learned carefully from the editor’s and reviewer’s comments, which are very valuable and very helpful for revising and improving our paper. After studying the critical comments, we have responded point by point and made corresponding changes in our manuscript. Our responses to the editor’s and reviewer’s comments are as follows:

Reply to Comments of Editors:

(I)  Please check that all references are relevant to the contents of the manuscript.

(II) Any revisions to the manuscript should be marked up using the “Track Changes” function if you are using MS Word/LaTeX, such that any changes can be easily viewed by the editors and reviewers.

(III) Please provide a cover letter to explain, point by point, the details of the revisions to the manuscript and your responses to the referees ‘comments.

(IV) If you found it impossible to address certain comments in the review reports, please include an explanation in your appeal.

(V) The revised version will be sent to the editors and reviewers.

Reply to Editors:

1: Please check that all references are relevant to the contents of the manuscript.

>>>Re: Thanks for your comments. We checked all references, which are relevant to the contents of the manuscript.

2: Any revisions to the manuscript should be marked up using the “Track Changes” function if you are using MS Word/LaTeX, such that any changes can be easily viewed by the editors and reviewers.

>>>Re: Thanks for your comments. We marked up all revisions to the manuscript using the “Track Changes”.

3: Please provide a cover letter to explain, point by point, the details of the revisions to the manuscript and your responses to the referees ‘comments.

>>>Re: We provided a cover letter to explain, point by point, the details of the revisions to the manuscript and our responses to the referees ‘comments.

4: If you found it impossible to address certain comments in the review reports, please include an explanation in your appeal.

>>>Re: Thanks for your suggestions. We have tried our best to respond to the comments from the reviewers point by point.

5: The revised version will be sent to the editors and reviewers.

>>>Re: Thanks.

Reply to Comments of Academic Editor:

This paper is not written according to the IJMS format. For example, the methods, results, and conclusions should not be written before the introduction. The reference does not also follow the IJMS format. Authors should follow the IJMS format before submitting the manuscript. The authors should rewrite according to the IJMS format and resubmit the manuscript again.

>>>Re: We are sorry for not writing according to the IJMS format. We have rewritten the manuscript according to the IJMS format.

For example, the abstract was revised as:“Simulated-daylight-photodynamic therapy (SD-PDT) maybe is an efficacious strategy for treating melanoma because it can overcome severe stinging pain, erythema, and edema during conventional PDT. However, the poor daylight response of existing common photosensitizers leads to unsatisfactory anti-tumor therapeutic effects and limits the development of daylight PDT. Hence, in this study, we utilized Ag nanoparticles to adjust the daylight response of TiO2 and then acquire efficient photochemical activity and the enhanced anti-tumor therapeutic effect of SD-PDT on melanoma. The synthesized Ag-doped TiO2 showed an optimal enhanced effect compared to Ag-core TiO2. Doping Ag into TiO2 produced a new shallow acceptor impurity level in the energy band structure, which expanded optical absorption in the range of 400–800 nm, and finally, improved the photodamage effect of TiO2 under SD irradiation. The plasmonic near-field distributions were enhanced due to the high refractive index of TiO2 at the Ag-TiO2 interface, and then, the amount of light captured by TiO2 was increased to induce the enhanced SD-PDT effect induced by Ag-core TiO2. Hence, Ag could effectively improve photochemical activity and SD-PDT effect of TiO2 through the change of the energy band structure. Generally, Ag-doped TiO2 is a promising photosensitizer agent for treating melanoma via SD-PDT.”

Reply to Comments of Reviewer 1:

REVIEWER 1 EVALUATION

Overview:

The article from Cuiping Yao et al. entitled “Improved simulated-daylight-photodynamic therapy and possible mechanism of Ag-modified TiO2 on Melanoma” deals with the preparation of Ag-doped TiO2 and Ag-core TiO2 nanoparticles for Simulated-daylight-photodynamic therapy (SD-PDT). The authors prepared these nanoparticles and characterized them by TEM, DRX and UV-Vis. They assessed the photodegradation of the nanoparticles using methylene blue and evaluated their phototoxicity and ROS generation under sunlight irradiation in cell lines. Then, the authors related the photophysical and structural properties by density functional studies. The aims of the work are clear and well-presented using several techniques/methods. Overall, the study deserves of publication in IJMS although some points shall be solved beforehand:

  • The authors shall revise the English. There are some typos along the text, i.e., “sliver” line 340.

>>>Re: Thanks for your constructive comments. We obtained help from a native English speaker to revise the typographical and language errors in the paper. The English language and grammar in the revised manuscript has been improved. And some typos have revised. For example:

“PDT efficiency relies on three primary factors: photosensitizer, light, and molecular oxygen, as per the PDT mechanism.”

“TiO2 is photoactive in the presence of UV light, which provides a basis for applying daylight PDT”

“The Ag nanoparticles were produced utilizing the seed growth method.”

“Nonetheless, the absorption spectrum of Ag-doped TiO2 displayed a prominent redshift (Figure 2A), which may be attributed to alterations in the energy band structure. The XRD findings demonstrated that the lattice structure of TiO2 remained unaltered following Ag doped into TiO2 (Figure 2B). In Ag-core TiO2, addition of sodium bicarbonate to the reaction mixture led to the formation of a TiO2 shell enveloping Ag nanoparticles.”

“Upon reaching a 2% Ag concentration, the degradation rate rose from 40% to 70%.”

“Ag proved to be highly effective in enhancing the photochemical activity and PDT effect of TiO2 when exposed to simulated daylight irradiation on melanoma. Thus, Ag-doped TiO2 exhibits great potential as a photosensitizer agent for treating melanoma with daylight PDT.”

  • The theoretical mechanistic section must be explained better since it is difficult to follow their analysis and conclusions.

>>>Re: Thanks for your constructive comments. We have rewritten the theoretical mechanistic section.

“The results of the experiment showed that modification with Ag could effectively enhance the photodamage effect of TiO2 under simulated daylight irradiation. Specifically, Ag-doped TiO2 had a more significant daylight PDT effect on melanoma. This might be due to an increase in the response of TiO2 to daylight facilitated by Ag. To confirm whether this mechanism was used, the changes in the optical absorption properties of TiO2 induced by doping with Ag were used based on the density functional theory using the CASTEP code. First, a supercell of TiO2 (contained 16 Ti atoms and 32 O atoms) and 2% Ag-doped TiO2 (contained 15 Ti atoms, 32 O atoms and 1 Ag atom) was constructed and used as the later calculation model (Figure 6A and 6B). And then the changes in the band structure and the density of the states of pure TiO2 and Ag-doped TiO2 were calculated. The energy of band gap of pure TiO2 was 3.325 eV, which then decreased to 3.14 eV when Ag was doped (Figure 6C and 6D). Additionally, compared to the band structure of pure TiO2, two new impurity energy levels were introduced, and one of them passed through the Fermi level, which indicated that it can act as the acceptor at the shallow impurity energy level as a bound state of a hole (Figure 6C and 6D). This shallow acceptor impurity level could enhance the separation of electron-hole pairs, produce free conduction holes, and decrease the recombination of photo-generated electrons and holes and decrease the energy required for the electrons to escape.

The computed results of the total density of states and partial density of states of pure TiO2 and Ag-doped TiO2 showed that the 4d-orbital electrons of Ag atoms were used for introducing the new impurity energy levels and producing the new electronic states through strong mixing with the p-orbital states of oxygen atoms (Figure 7). Hence, these two impurity energy levels of Ag-doped TiO2 mainly occurred due to doping with Ag. Generally, the doped Ag atom have influenced on the TiO2 energy structure and induced the band-gap narrowing. Ag is the main reason to induce the increase of the daylight response of Ag-doped TiO2. Ag-doped TiO2 expanded optical absorption in the range of 400–800 nm and improved the photodamage effect of TiO2 under daylight irradiation.

First-Principles analysis can be used to analyze the properties of materials with a crystal structure but not of materials with a core-shell structure. To determine the cause of the improvement in the SD-PDT effect of TiO2 induced by Ag-core TiO2, discrete dipole approximation simulations were studied because these changes might be induced by the localized surface plasmon resonance enhancement effect. First, a series of Ag-core TiO2 with a constant Ag core and TiO2 shells of different thicknesses (0 nm, 5 nm, 10 nm, 15 nm, and 20 nm) were calculated. As shown in Figure 8A, the results showed the absorption spectrum has an obvious red shift and a significant decrease in the intensity with an increase in the thicknesses of the TiO2 shell. These results matched the absorption spectrum data we measured. The plasmonic near-field distribution showed a strong change in the Ag-core TiO2 (Figure 8B). The electric near-field intensities were considerably enhanced due to the high refractive index of TiO2 at the Ag- TiO2 interface. With the increase in the electric near-field intensity, the amount of light captured by TiO2 and the PDT effect increased. The relationship between the field enhancement effect and the thickness of the TiO2 shell results showed that the field enhancement effect gradually weakened and the ability of the TiO2 shell to capture light decreased with an increase in the thickness of the shell, which occurred probably because the shell affected the movement of photo-generated electrons and holes (Figure 8C). Hence, the Ag-core TiO2 with a 5-nm thick shell had the highest photocatalytic efficiency.

  • The ROS generation study is not well-suited since the authors have to quantify the fluorescence emission somehow. Just the observation of fluorescence in a well-plate full of cells is not significant. The authors shall re-plan and re-do the assay.

>>>Re: Thanks for your constructive comments. It is very important to quantify the fluorescence intensity of ROS concentration after treated by Ag-doped TiO2, Ag-core TiO2, the synthesized TiO2 and P25 under a simulated sunlight Xenon lamp irradiation. Hence, we measured and analyzed the fluorescence intensity of DCFH-DA fluorescence probe by fluorescence spectrophotometer to detect the concentration of ROS in cells. The methodology, results and Figure have added and revised also, as following:

“To quantify the ability of generation of ROS, the fluorescence intensity of DCFH-DA was measured by fluorescence spectrophotometer to detect the concentration of ROS in cells after treated by Ag-doped TiO2, Ag-core TiO2, the synthesized TiO2 and P25 under a simulated sunlight Xenon lamp irradiation. The treated cells under the same conditions above were harvested, incubated with 10 μmol/L DCFH-DA for 10 min at 37 °C in complete darkness, and then centrifuged, washed with PBS and measured using a fluorescence spectrophotometer under an excitation of 488nm light. In order to reveal the role of ROS more directly in simulated daylight-PDT induced by Ag-modified TiO2, inhibition testes were measured by quenching agent of ROS (histidine). After treated with the different agents containing TiO2,the cells were treated with 20 mM histidine for 30 min, washed with PBS and then irradiated with sunlight Xenon lamp and detected the cell activity by CCK-8 analysis as same as before.”

“To determine the simulated daylight PDT effect induced by Ag-modified TiO2, the ability to generate ROS was evaluated by the DCFH-DA probe. The fluorescence imaging results showed that ROS was generated after induction by P25, the synthesized TiO2 and Ag-modified TiO2, and the induction was higher when Ag-modified TiO2 was used (Figure 4A). The fluorescence intensity of DCFH-DA increased obviously in Ag-modified TiO2 (Figure 4B). Compared with P25, Ag-core TiO2 and Ag-doped TiO2 resulted in significant increases at average values of 1.75-fold and 1.95-fold respectively. Hence, the ability of ROS generations induced by Ag-doped TiO2 were higher than those induced by Ag-core TiO2. The inhibitory activity induced by all reagents containing TiO2 could be effectively weakened by using a quenching agent of ROS. After treated with P25, the synthesized TiO2, Ag-core TiO2 and Ag-doped TiO2, and 20mM histidine, the cell activities increased from 67.6%, 60%, 31.6% and 24.5% to 85.3%, 80.8%, 60.2% and 55.6% respectively. This revealed that the degree of inhibition induced by Ag-modified TiO2 was higher than that induced by the synthesized TiO2 or P25 (Figure 4C). Hence, our findings showed that Ag-modified TiO2, especially Ag-doped TiO2, efficiently improved the simulated daylight PDT by enhancing the ability to generate ROS.”

  • The methodology is not explained. It shall be completed, in this regards cell viability assay is not possible to understand. There is some part of the text missed.

>>>Re: We are sorry for not writing according to the IJMS format. Hence, the methodology was showed in Part 3 of the manuscript. We have rewritten the manuscript according to the IJMS format and the methodology was showed in Part 2 of the revised manuscript, which included the cell viability assay.

  • Why do the authors used A375 cells?

>>>Re: Photodynamic therapy is an effective therapeutic strategy for skin disease in clinical therapy. However, it is often accompanied by severe stinging pain, erythema, and edema. In 2008, daylight PDT was first introduced as a less painful outdoors alternative to conventional PDT, with similar clinical effectiveness. Hence, daylight PDT maybe is an effective therapeutic strategy for skin disease. Melanoma is a commonly occurring severe skin malignancy. Its incidence is ever-increasing. However, the daylight PDT anti-tumor therapeutic effect is very limited on melanoma. It maybe caused by the poor daylight response of existing common photosensitizers. Hence, in this study, we are trying to find a strategy that can improve the daylight response and then increase the daylight PDT effect on melanoma. A375 cells are a melanoma cell line and often used as an in vitro model in the study of melanoma. Hence, in this study, we used A375 cells to study Ag-modified TiO2 daylight PDT effect on melanoma.

(6) Some parts of the text are duplicated, for instance lines 176-180 and lines 42-46. The authors shall be rewritten these parts.

>>>Re: Thanks for your constructive comments. We have rewritten the manuscript and revised the duplicated parts. For example:

We rewritten the part that duplicated with lines 42-46 to “During PDT, generated ROS can lead to damage cellular components and then induce cell death. Hence, the ability to generate ROS determines the PDT effect.”

We rewritten the duplicated sentences (lines 176-180) to “The cytotoxicity assay and phototoxicity of Ag-modified TiO2 through CCK-8 assay. A: Cell viability without irradiation-the cytotoxicity assay of Ag-modified TiO2 compared with P25 and the synthesized TiO2. B: Cell viability after irradiation by daylight-the phototoxicity assay of Ag-modified TiO2 compared with P25 and the synthesized TiO2 with the different concentration of TiO2. C: Cell viability after different irradiation dosage-the phototoxicity assay of Ag-modified TiO2 compared with P25 and the synthesized TiO2. *, P< 0.05, represent statistically difference between P25, the synthesized TiO2, Ag-core TiO2, Ag-doped TiO2 group and control group.”

Once again, thank you and all the reviewers for the kind advice.

Reviewer 2 Report

Manuscript: ijms-2299785 presents a study with scientific significance in the fields of biomedicine and medicinal chemistry and the results are of practical importance for melanoma therapeutics although further and detailed clinical investigations are necessary. The manuscript needs major revision before publication concerning the following issues:

1.       Please explain weather the PDL is simulated or stimulated as in the title and conclusions the authors mention simulated, while within the abstract and the text, e.g. Lines 167, 182, 190, 305 it is written stimulated.

2.       The last part of the Introduction section from line 79 to line 93 has to be removed as the authors presented a resume of the study, which is not necessary in the introduction. Instead the main goal of the study has to be given.

3.       2.1. Synthesis and characterization of Ag-modified TiO2 with the different structure – remove “the”.

4.       Line 274 - The obtained dried product was fully milled.

5.       The types and producers of all apparatuses have to be given in Materials and methods

6.       Subsection Reagents has to be included in Section Materials and methods.

7.       The sentence in Lines 286-287 “The purified silver nanoparticles could be obtained.” needs revision.

8.       Line 290 – don not capitalize “butyl””

9.       Line 310 – after incubation

10.   Lines 320-321 – present the calculation as a formula and explain the abbreviations (OD – optical density)

11.   Line 328 - for twice.

12.   Lines 340-341 the sentence needs revision.

13.   Line 100 – What does P25 mean?

14.   Figure 1 H – What does EDS stand for?

15.   Figure 1 L is not clear. It is difficult to understand what it represents!

16.   Lines 134 – 136: The black curve on Figure 2A displays decreasing MB concentration with time, which obviously is an indication of the dye degradation. Could you explain this discrepancy?

17.   Lines 176-180 – please add suitable references!

18.   Statistical analyses and statistical significance of the results are missing.

19.   According to the Guidelines of the journal for researches Involving Cell Lines: Methods sections for submissions reporting on research with cell lines should state the origin of any cell lines. For established cell lines the provenance should be stated and references must also be given to either a published paper or to a commercial source. If previously unpublished de novo cell lines were used, including those gifted from another laboratory, details of institutional review board or ethics committee approval must be given, and confirmation of written informed consent must be provided if the line is of human origin.

Author Response

Revised manuscript submitted to International Journal of Molecular Science

Manuscript ID: IJMS-2299785

Title: Improved simulated-daylight-photodynamic therapy and possible mechanism of Ag-modified TiO2 on Melanoma

Authors: Jing Xin, Jing Wang, Yuanping Yao, Sijia Wang, Zhenxi Zhang and Cuiping Yao*

Dear Editors and Reviewers,

  Thank you very much for your evaluation and comments from the reviewers for our manuscript. We have learned carefully from the editor’s and reviewer’s comments, which are very valuable and very helpful for revising and improving our paper. After studying the critical comments, we have responded point by point and made corresponding changes in our manuscript. Our responses to the editor’s and reviewer’s comments are as follows:

Reply to Comments of Editors:

(I)  Please check that all references are relevant to the contents of the manuscript.

(II) Any revisions to the manuscript should be marked up using the “Track Changes” function if you are using MS Word/LaTeX, such that any changes can be easily viewed by the editors and reviewers.

(III) Please provide a cover letter to explain, point by point, the details of the revisions to the manuscript and your responses to the referees ‘comments.

(IV) If you found it impossible to address certain comments in the review reports, please include an explanation in your appeal.

(V) The revised version will be sent to the editors and reviewers.

Reply to Editors:

1: Please check that all references are relevant to the contents of the manuscript.

>>>Re: Thanks for your comments. We checked all references, which are relevant to the contents of the manuscript.

2: Any revisions to the manuscript should be marked up using the “Track Changes” function if you are using MS Word/LaTeX, such that any changes can be easily viewed by the editors and reviewers.

>>>Re: Thanks for your comments. We marked up all revisions to the manuscript using the “Track Changes”.

3: Please provide a cover letter to explain, point by point, the details of the revisions to the manuscript and your responses to the referees ‘comments.

>>>Re: We provided a cover letter to explain, point by point, the details of the revisions to the manuscript and our responses to the referees ‘comments.

4: If you found it impossible to address certain comments in the review reports, please include an explanation in your appeal.

>>>Re: Thanks for your suggestions. We have tried our best to respond to the comments from the reviewers point by point.

5: The revised version will be sent to the editors and reviewers.

>>>Re: Thanks.

Reply to Comments of Academic Editor:

This paper is not written according to the IJMS format. For example, the methods, results, and conclusions should not be written before the introduction. The reference does not also follow the IJMS format. Authors should follow the IJMS format before submitting the manuscript. The authors should rewrite according to the IJMS format and resubmit the manuscript again.

>>>Re: We are sorry for not writing according to the IJMS format. We have rewritten the manuscript according to the IJMS format.

For example, the abstract was revised as:“Simulated-daylight-photodynamic therapy (SD-PDT) maybe is an efficacious strategy for treating melanoma because it can overcome severe stinging pain, erythema, and edema during conventional PDT. However, the poor daylight response of existing common photosensitizers leads to unsatisfactory anti-tumor therapeutic effects and limits the development of daylight PDT. Hence, in this study, we utilized Ag nanoparticles to adjust the daylight response of TiO2 and then acquire efficient photochemical activity and the enhanced anti-tumor therapeutic effect of SD-PDT on melanoma. The synthesized Ag-doped TiO2 showed an optimal enhanced effect compared to Ag-core TiO2. Doping Ag into TiO2 produced a new shallow acceptor impurity level in the energy band structure, which expanded optical absorption in the range of 400–800 nm, and finally, improved the photodamage effect of TiO2 under SD irradiation. The plasmonic near-field distributions were enhanced due to the high refractive index of TiO2 at the Ag-TiO2 interface, and then, the amount of light captured by TiO2 was increased to induce the enhanced SD-PDT effect induced by Ag-core TiO2. Hence, Ag could effectively improve photochemical activity and SD-PDT effect of TiO2 through the change of the energy band structure. Generally, Ag-doped TiO2 is a promising photosensitizer agent for treating melanoma via SD-PDT.”

Reply to Comments of Reviewer2:

REVIEWER 2 EVALUATION

Manuscript: ijms-2299785 presents a study with scientific significance in the fields of biomedicine and medicinal chemistry and the results are of practical importance for melanoma therapeutics although further and detailed clinical investigations are necessary. The manuscript needs major revision before publication concerning the following issues:

  1. Please explain weather the PDL is simulated or stimulated as in the title and conclusions the authors mention simulated, while within the abstract and the text, e.g. Lines 167, 182, 190, 305 it is written stimulated.

>>>Re: Sorry for this mistake. In this study, we researched simulated-daylight-PDT. We have revised stimulated to simulated in our manuscript.

  1. The last part of the Introduction section from line 79 to line 93 has to be removed as the authors presented a resume of the study, which is not necessary in the introduction. Instead the main goal of the study has to be given.

>>>Re: Many thanks for your suggestion. The introduction section from line 79 to line 93 was revised as:“To shift the TiO2 absorption spectrum to the visible region to expand the daylight response range, several approaches have been proposed. Among them, doping with the metal ions using transition metal or non-metal ions to change the optoelectronic features of TiO2 could significantly shift the optical response of TiO2. In addition, modifying with plasmonic metallic nanoparticles to combinate the photocatalytic properties of TiO2 and the optical properties of plasmonic nanoparticles could extent the photocatalytic activity of TiO2 from UV light to visible or even to the NIR range of radiation. Among all metallic materials, silver (Ag) exhibits the most interesting physical properties and unique optical properties. Hence, in this study, the Ag-modified TiO2 nanomaterials with different structures (Ag-doped TiO2 and Ag-core TiO2) were synthesized to improve the limited SD-PDT effect on melanoma by increasing the daylight response. The improvement in the photochemical activity and the therapeutic effect of PDT were compared and the possible mechanisms were theoretically studied.”

  1. 2.1. Synthesis and characterization of Ag-modified TiO2 with the different structure – remove “the”.

>>>Re: Thanks and done.

  1. Line 274 - The obtained dried product was fully milled.

>>>Re: Thanks and done.

  1. The types and producers of all apparatuses have to be given in Materials and methods

>>>Re: Thanks for your constructive comments. We have added the types and producers of all apparatuses in Materials and methods. For example:

“The morphologies of Ag-doped TiO2 and Ag-core TiO2 were observed using transmission electron microscopy (TEM; JEM-2100, JEOL, Tokyo, Japan). Absorption spectra of Ag-doped TiO2 and Ag-core TiO2 were recorded using an ultraviolet–visible spectrophotometer (V-550 UV/VIS, JASCO, Tokyo, Japan). X-ray diffraction was done using x-ray diffractometer (X’pert Powder, PANalytical B.V. Netherlands). Energy dispersive spectrometer was used to observe the distribution pattern of various elements (Ag, Ti and O) using TEM-EDS (JEM-2100 Plus, JEOL Corporation, Japan), operating with an accelerating voltage of 200 kV.”

  1. Subsection Reagents has to be included in Section Materials and methods.

>>>Re: Thanks for your good suggestion. We have added the subsection reagents in Materials and methods.

“Silver nitrate (AgNO3), tetra butyl titanate and P25 (TiO2 nanoparticles) were purchased from Sigma. TiCl3 was purchased from Aladdin. Absolute ethanol, sodium citrate, solidum borohydride (NaBH4), butyl alcohol, sodium bicarbonate (NaHCO3) and N-Butanol were purchased from Tianjin Fuyu Chemical Co., Ltd and Tianjin Tianli Chemical Reagent Co., Ltd (China). Cell Counting Kit (CCK-8) was purchased from Dojindo (Japan). The DCFH-DA was purchased from Beyotime Company (China). Human melanoma cell line A375 was obtained from the Cell Bank of Chinese Academy of Sciences (Shanghai, China). The A375 cells were cultured in DMEM medium (HyClone) supplemented with 10% fetal bovine serum (HyClone) and 1% penicillin/streptomycinin a humidified incubator at 37 °C with 5% CO2.

  1. The sentence in Lines 286-287 “The purified silver nanoparticles could be obtained.” needs revision.

>>>Re: Thanks. We have revised “The purified silver nanoparticles could be obtained.” to “The purified Ag nanoparticles was yielded.”

  1. Line 290 – don not capitalize “butyl””

>>>Re: Thanks and done.

  1. Line 310 – after incubation

>>>Re: Thanks and done.

  1. Lines 320-321 – present the calculation as a formula and explain the abbreviations (OD – optical density)

>>>Re: Thanks and done.

  1. Line 328 - for twice.

>>>Re: Thanks and done.

  1. Lines 340-341 the sentence needs revision.

>>>Re: We have revised“This electric field enhancement could be verified through discrete dipole approximation simulations”to“This electric field enhancement factors can be quantified and analyzed using simulations based on discrete dipole approximation.”

  1. Line 100 – What does P25 mean?

>>>Re: P25 is commercialized TiO2 nanoparticles purchased from Sigma. We have revised this sentence to“The TEM results of the synthesized TiO2 were comparable to P25 (the commercialized TiO2 nanoparticles) purchased from Sigma.”

  1. Figure 1 H – What does EDS stand for?

>>>Re: DES stands for Energy Dispersive Spectrometer. We used TEM-EDS (JEM-2100Plus, JEOL Corporation, Japan) to observe the distribution pattern of various elements (Ag, Ti and O), operating with an accelerating voltage of 200 kV. We have added it in the manuscript.

  1. Figure 1 L is not clear. It is difficult to understand what it represents!

>>>Re: TiO2 naturally occurs in three distinct crystalline phases with different physical and chemical properties: brookite (orthorhombic crystal structure), anatase (tetragonal crystal structure), and rutile (tetragonal crystal structure). Under ambient conditions, bulk rutile is thermodynamically stable, whereas anatase and brookite are thermodynamically metastable. Thus, these phases can be accessed through thermally driven phase transformations, as may occur during calcination. Among the three phases, anatase and rutile are widely used for photocatalytic applications because of their facile synthesis. In contrast, brookite has been rarely investigated as a photocatalyst because it is not readily accessible in its pure form. Hence, in this study, we synthesized TiO2 and calcined at 450°C for 2h to obtain anatase TiO2. Figure 1L was used to observe the crystal lattice structure of TiO2 after calcination. We have revised the captions on Figures 1L to “The crystal lattice structure of TiO2 in Ag-core TiO2 observed using transmission electron microscopy image with high resolution after the calcination treatment.”

  1. Lines 134 – 136: The black curve on Figure 2A displays decreasing MB concentration with time, which obviously is an indication of the dye degradation. Could you explain this discrepancy?

>>>Re: Thanks for your comments. The black curve on Figure 2A displays photocatalytic activity of our synthesized TiO2 through MB dye degradation. TiO2 itself has a good photocatalytic activity under ultraviolet radiation. In our study, the absorption of the our synthesized TiO2 in the range of 400-800 nm is higher than that of P25-the commercialized TiO2 nanoparticles. Hence, the photocatalytic activity of our synthesized TiO2 displayed decreasing obviously MB concentration with time and that is better than that of P25 under simulated daylight irradiation.

  1. Lines 176-180 – please add suitable references!

>>>Re: Thanks and done.

  1. Statistical analyses and statistical significance of the results are missing.

>>>Re: Thanks and done.

  1. According to the Guidelines of the journal for researches Involving Cell Lines: Methods sections for submissions reporting on research with cell lines should state the origin of any cell lines. For established cell lines the provenance should be stated and references must also be given to either a published paper or to a commercial source. If previously unpublished de novo cell lines were used, including those gifted from another laboratory, details of institutional review board or ethics committee approval must be given, and confirmation of written informed consent must be provided if the line is of human origin.

>>>Re: Thanks for your suggestion. We have stated the origin of A375 cell line and added the reference in Regents and cell line. “Human melanoma cell line A375 was obtained from the Cell Bank of Chinese Academy of Sciences (Shanghai, China) [38].”

  1. Yu, B.; Wang, Y.; Yu, X.; Zhang, H.; Zhu, J.; Wang, C.; Chen, F.; Liu, C.; Wang, J.; Zhu, H. Cuprous oxide nanoparticle-inhibited melanoma progress by targeting melanoma stem cells. Int J Nanomedicine 2017, 12, 2553-2567, doi:10.2147/IJN.S130753.

Once again, thank you and all the reviewers for the kind advice.

Reviewer 3 Report

1. Figure 1 will be more understandable if it is reorganized and presented in 2 Figures:

Figure 1- Transmission electron microscopy image of Ag-modified TiO2: Figures (A,B,E,F, I,J, K,L) with the designations respectively changed to (A,B,C,D,E,F,G,H) ;

Figure 2 - Properties of Ag-modified TiO2:
Figures (C,D,G,H ) with the designations respectively changed to (A,B,C,D)

2.
The scale in Figure 1 (I,J) - does not correspond to the size of Ag-core TiO2 with 20 nm thickness of TiO2 shell and TiO2 agglomerated - check and replace.

3. The C
aptions on Figures 1 (I,J) should be changed to the same inscriptions:

Figure 1(I) - image of Ag-core TiO2 with 20 nm thickness of TiO2 shell (sodium bicarbonate at 3 mL);

Figure 1(J) image of Ag-core TiO2 with shell of TiO2 agglomerated (sodium bicarbonate at 1.5 mL).

4. Figure 1(K)- no scale - add.

5. Line 149: replace the phrase "we determined the cell viability" with "we determined the viability of A375 line cells ( human melanoma cell line)......

Line 150: CCK-8 - decipher (Cell Counting Kit-8 - allows sensitive colorimetric assays for the determination of cell viability in cell proliferation and cytotoxicity assays).

6. Line 154-155- "As shown in Figure 3B, after irradiation", add "by daylight"

7. The
Caption for Figure 3 would be clearer if changed to:

A- Cell viability without iradiation - the cytotoxicity assay of Ag-modified TiO2 compared with P25 and the synthesized TiO2

B- Cell viability after irradiation by daylight - the phototoxicity assay of Ag-modified TiO2 compared with P25 and the synthesized TiO2 with the different concentration of TiO2

C - Cell viability after different irradiation dosage - the phototoxicity assay of Ag-modified TiO2 compared with P25 and the synthesized TiO2

8.
Line 183: to "DCFH-DA probe" add - "the most widely used probe for detecting intracellular H2O2 and oxidative stress".

9. Correct the designations of the Figure 4:
- give the name of the cell line;
- correct discrepancies in the labeling on the figures (Ag@TiO2 Ag/TiO2 ) and in the
figure designations (Ag-core TiO2 and Ag-doped TiO2;
- indicate, what is represented in the right, center, and left columns.

10. Figure 5 - indicate, what is indicated on the abscissa axis and what do the dashed red and blue lines mean?

11. Include in article the "Discussion" section. Is your work pioneering, advantages of silver compared to other ligating materials, example, such as gold, advantages and disadvantages of the work, future plans, etc.

12.
Line 306: to "effect on melanoma cells" add "A375 line".

Author Response

Revised manuscript submitted to International Journal of Molecular Science

Manuscript ID: IJMS-2299785

Title: Improved simulated-daylight-photodynamic therapy and possible mechanism of Ag-modified TiO2 on Melanoma

Authors: Jing Xin, Jing Wang, Yuanping Yao, Sijia Wang, Zhenxi Zhang and Cuiping Yao*

Dear Editors and Reviewers,

  Thank you very much for your evaluation and comments from the reviewers for our manuscript. We have learned carefully from the editor’s and reviewer’s comments, which are very valuable and very helpful for revising and improving our paper. After studying the critical comments, we have responded point by point and made corresponding changes in our manuscript. Our responses to the editor’s and reviewer’s comments are as follows:

Reply to Comments of Editors:

(I)  Please check that all references are relevant to the contents of the manuscript.

(II) Any revisions to the manuscript should be marked up using the “Track Changes” function if you are using MS Word/LaTeX, such that any changes can be easily viewed by the editors and reviewers.

(III) Please provide a cover letter to explain, point by point, the details of the revisions to the manuscript and your responses to the referees ‘comments.

(IV) If you found it impossible to address certain comments in the review reports, please include an explanation in your appeal.

(V) The revised version will be sent to the editors and reviewers.

Reply to Editors:

1: Please check that all references are relevant to the contents of the manuscript.

>>>Re: Thanks for your comments. We checked all references, which are relevant to the contents of the manuscript.

2: Any revisions to the manuscript should be marked up using the “Track Changes” function if you are using MS Word/LaTeX, such that any changes can be easily viewed by the editors and reviewers.

>>>Re: Thanks for your comments. We marked up all revisions to the manuscript using the “Track Changes”.

3: Please provide a cover letter to explain, point by point, the details of the revisions to the manuscript and your responses to the referees ‘comments.

>>>Re: We provided a cover letter to explain, point by point, the details of the revisions to the manuscript and our responses to the referees ‘comments.

4: If you found it impossible to address certain comments in the review reports, please include an explanation in your appeal.

>>>Re: Thanks for your suggestions. We have tried our best to respond to the comments from the reviewers point by point.

5: The revised version will be sent to the editors and reviewers.

>>>Re: Thanks.

Reply to Comments of Academic Editor:

This paper is not written according to the IJMS format. For example, the methods, results, and conclusions should not be written before the introduction. The reference does not also follow the IJMS format. Authors should follow the IJMS format before submitting the manuscript. The authors should rewrite according to the IJMS format and resubmit the manuscript again.

>>>Re: We are sorry for not writing according to the IJMS format. We have rewritten the manuscript according to the IJMS format.

For example, the abstract was revised as:“Simulated-daylight-photodynamic therapy (SD-PDT) maybe is an efficacious strategy for treating melanoma because it can overcome severe stinging pain, erythema, and edema during conventional PDT. However, the poor daylight response of existing common photosensitizers leads to unsatisfactory anti-tumor therapeutic effects and limits the development of daylight PDT. Hence, in this study, we utilized Ag nanoparticles to adjust the daylight response of TiO2 and then acquire efficient photochemical activity and the enhanced anti-tumor therapeutic effect of SD-PDT on melanoma. The synthesized Ag-doped TiO2 showed an optimal enhanced effect compared to Ag-core TiO2. Doping Ag into TiO2 produced a new shallow acceptor impurity level in the energy band structure, which expanded optical absorption in the range of 400–800 nm, and finally, improved the photodamage effect of TiO2 under SD irradiation. The plasmonic near-field distributions were enhanced due to the high refractive index of TiO2 at the Ag-TiO2 interface, and then, the amount of light captured by TiO2 was increased to induce the enhanced SD-PDT effect induced by Ag-core TiO2. Hence, Ag could effectively improve photochemical activity and SD-PDT effect of TiO2 through the change of the energy band structure. Generally, Ag-doped TiO2 is a promising photosensitizer agent for treating melanoma via SD-PDT.”

Reply to Comments of Reviewer3:

REVIEWER 3 EVALUATION

  1. Figure 1 will be more understandable if it is reorganized and presented in 2 Figures:

Figure 1- Transmission electron microscopy image of Ag-modified TiO2: Figures (A,B,E,F, I,J, K,L) with the designations respectively changed to (A,B,C,D,E,F,G,H) ;
Figure 2 - Properties of Ag-modified TiO2:
Figures (C,D,G,H ) with the designations respectively changed to (A,B,C,D)
>>>Re: Thanks for your good suggestions. We have reorganized and presented Figure 1 in 2 Figures.

  1. The scale in Figure 1 (I,J) - does not correspond to the size of Ag-core TiO2with 20 nm thickness of TiO2 shell and TiO2 agglomerated - check and replace.
    >>>Re: Sorry for this mistake. We have checked and replaced the shell size of Ag-core TiO2 to about 18.7 nm. And we have changed the TEM image (H), which is better exhibited TiO2 agglomeration.

  2. The Captions on Figures 1 (I,J) should be changed to the same inscriptions:

    Figure 1(I) - image of Ag-core TiO2with 20 nm thickness of TiO2shell (sodium bicarbonate at 3 mL);

    Figure 1(J) image of Ag-core TiO2 with shell of TiO2 agglomerated (sodium bicarbonate at 1.5 mL).
    >>>Re: Thanks and done.
    4. Figure 1(K)- no scale - add.
    >>>Re: Thanks and done.
    5. Line 149: replace the phrase "we determined the cell viability" with "we determined the viability of A375 line cells ( human melanoma cell line)......
    >>>Re: Thanks and done.
    Line 150: CCK-8 - decipher (Cell Counting Kit-8 - allows sensitive colorimetric assays for the determination of cell viability in cell proliferation and cytotoxicity assays).
    >>>Re: Thanks and added.
    6. Line 154-155- "As shown in Figure 3B, after irradiation", add "by daylight"
    >>>Re: Thanks and added.
    7. The Caption for Figure 3 would be clearer if changed to:

A- Cell viability without iradiation - the cytotoxicity assay of Ag-modified TiO2 compared with P25 and the synthesized TiO2

B- Cell viability after irradiation by daylight - the phototoxicity assay of Ag-modified TiO2 compared with P25 and the synthesized TiO2 with the different concentration of TiO2

C - Cell viability after different irradiation dosage - the phototoxicity assay of Ag-modified TiO2 compared with P25 and the synthesized TiO2
>>>Re: Thanks and changed.
8. Line 183: to "DCFH-DA probe" add - "the most widely used probe for detecting intracellular H2O2 and oxidative stress".
>>>Re: Thanks and added.
9. Correct the designations of the Figure 4:
- give the name of the cell line;
- correct discrepancies in the labeling on the figures (Ag@TiO2 Ag/TiO2 ) and in the figure designations (Ag-core TiO2 and Ag-doped TiO2;
- indicate, what is represented in the right, center, and left columns.
>>>Re: Thanks and done.
10. Figure 5 - indicate, what is indicated on the abscissa axis and what do the dashed red and blue lines mean?
>>>Re: The abscissa axis in Figure 5 indicated Brillouin zone for tetragonal structure of TiO2. The dashed red lines (energy zero) represent the valence-band maximum. The blue lines mean the minimum band gap at the G point. We have added the indication in Figure 5.

Brillouin zone for tetragonal structure of TiO2.

  1. Include in article the "Discussion" section. Is your work pioneering, advantages of silver compared to other ligating materials, example, such as gold, advantages and disadvantages of the work, future plans, etc.

>>>Re: Thanks for your comments. We have added the“Discussion”in the revised manuscript.

“Photodynamic therapy is an effective therapeutic strategy for skin disease in clinical therapy [42]. However, it is often accompanied by severe adverse effects during the treatment. The large number of patients are unable to continuing the treatment due to these adverse effects. In 2008, daylight-PDT was first introduced as a less painful outdoors alternative to conventional PDT, with similar clinical effectiveness [43]. But daylight-PDT efficacy often dependent on weather conditions. For example, in the U.K., daylight-PDT is practical between the months of March, April to October, September when the temperature is above 10°C in the day (from 9:00 to 18:00) and the fluence rate reaches 130 W/m2 [44]. In addition, to avoid the patient exposes to harmful wavelengths of ultraviolet radiation during daylight-PDT, the organic sunscreens should be used to prevent sun damage. To provide a controlled daylight-PDT environmental setting and remove the disadvantage of exposure to harmful ultraviolet radiation, SD-PDT has been investigated using an indoor daylight simulating lamp. Wulf and co-workers reported that four different lamp candidates (18 W red, 140 W red, and 50 W white light‐emitting diode lamps and halogen lamps from 250 W slide projectors as well as 400 W overhead projectors for SD-PDT were able to photo bleach PPIX photosensitizer completely [43]. Calzavara-Pinto et.al. revealed that SD-PDT using a lamp with output confined in the red waveband (6305nm) and a polychromatic white LED lamp (400-700 nm) can represent a valid therapeutic method for Actinic cheilitis [45]. In our study, the SD-PDT effect can be obtained under the irradiation of the sunlight Xenon lamp with the emission spectral range of 380nm to 700nm. Hence, the sunlight Xenon lamp (380-700 nm) is also a useful lamp candidate for SD-PDT. However, in our study, we did not evaluate and compare the SD-PDT effect of Ag-modified TiO2 under other lamp sources. In the further, more detailed comparative research may be needed to obtain better SD-PDT anti-tumor therapeutic effect.

In this study, the investigation of the improving strategy for increasing simulating daylight response of existing photosensitizers is the main purpose of the research that we want. TiO2 is a potent oxygen radical generator. However, it is limited in SD-PDT by the necessity to use ultraviolet irradiation of low tissue penetration and harmful impact on the human body. To maximize the visible light absorption of TiO2, inorganic compounds was usually doped to the TiO2 during their preparation because this process can narrow the bandgap in the TiO2 nanoparticle’s structure and decrease the necessary activation energy. Among these inorganic compounds, the noble metals (such as gold (Au), silver (Ag), platinum (Pt), palladium (Pd)) were used to doped with TiO2 one after another [33]. All absorption ranges of TiO2 were shifted to longer wavelengths and enhanced photocatalytic activities under visible light were obtained to different degree after doping. However, compared with other used noble metals, Ag has been regarded as a better candidate due to its higher catalytic activity and ROS generation ability [45-46]. Hence, Ag-doped TiO2 may be suitable for daylight PDT or SD-PDT. Unfortunately, there are few study reports that Ag-doped TiO2 is used to daylight PDT or SD-PDT. However, Alshamsan et.al. revealed that Ag-doped TiO2 have potential to selectively kill cancer cells while sparing normal cells through ROS generation in HepG2 (human liver cancer cell line) [47]. It gave us reason to do the research on evaluate SD-PDT effect of Ag-doped TiO2 on melanoma. In this study, the results showed that the limited photochemical activity and SD-PDT effect of TiO2 could be improve significantly through doping Ag to the TiO2. In addition, our results showed that the degree of the improvement photochemical activity was independent of the concentration introduction of Ag into TiO2. This may be caused by the synthesized Ag-doped TiO2 complex has different light response with different concentration of Ag under simulated daylight irradiation. This change in light response was not entirely dependent on the doped Ag concentration. Lu and co-workers measured Ag-doped TiO2 photocatalysts with different concentration Ag (1%-5%) and applied it to cement mortar in their previous study. They reported that 2% Ag-doped TiO2 had the highest photocatalytic activity under ultraviolet radiation and 5% Ag-doped TiO2 had the highest photocatalytic activity under solar light [46]. Hence, the introduction of the different concentration of Ag into TiO2 may cause the different change of light response. Generally, in our study, Ag-doped TiO2 within a certain concentration Ag efficiently improved TiO2 photochemical activity compared with TiO2.

Besides Ag-doped TiO2 complex, TiO2-coated Ag nanoparticle has found applications in many fields because which can combine the surface plasmon resonance properties of Ag core and the photoactivity of the TiO2 shell [48]. The tunable optical properties can be obtained through change the ratio of core radius and shell thickness. Hence, Ag-core TiO2 was also usually used to increase the optical absorption of TiO2 and extend its absorption region to the visible light. Like Ag-doped TiO2, there are few study reports that Ag-core TiO2 is used to daylight PDT or SD-PDT. And the improvement photocatalysts effect was not compared directly between Ag-doped TiO2 and Ag-core TiO2 in other study. To find a better the improving strategy for increasing simulating daylight response of TiO2 photosensitizer, the improvement in the photocatalytic activity and SD-PDT effect induced by Ag-doped TiO2 and Ag-coreTiO2 was compared. The results showed that the described TiO2 modification method based on Ag significantly increased the photochemical properties of TiO2. The synthesized Ag-doped TiO2 was found to be a promising agent for treating melanoma using daylight PDT, and doping Ag to TiO2 is optimal enhanced strategy.

Several studies revealed that the introduction of Ag into TiO2 improves TiO2 photochemical activity due to two mechanisms. (1): Ag can act as an electron acceptor to increase separation efficiency of photogenerated electron-hole pair because its Fermi level is below the conduction band of TiO2. (2): Generation local surface plasmon resonance effect to extend the visible light absorption range and increase photocatalytic efficiency of TiO2 [49]. Hence, in this study, the First-Principles analysis was performed for Ag-doped TiO2 and the Discrete Dipole approximation for Ag-core TiO2 was calculated. The results showed that a new shallow acceptor impurity level appeared in the energy band structure of Ag-doped TiO2, which decreased the recombination of photo-generated electrons and holes and the energy needed for the excitation of electrons. This expanded the light response range of TiO2 and made it more responsive to sunlight. A strong field enhancement effect was obtained at the interface between the TiO2 shell and the Ag core of Ag-core TiO2, which increased the amount of light captured by TiO2 and improved the photochemical activity. These are consistent with the previous described mechanism. These further confirm the reliability of our research on this the improving strategy for increasing simulating daylight response of TiO2 photosensitizer. Hence, Ag-doped TiO2 is a promising photosensitizer agent for treating melanoma with daylight PDT.”

  1. Line 306: to "effect on melanoma cells" add "A375 line".

>>>Re: Thanks and added.

Once again, thank you and all the reviewers for the kind advice.

Round 2

Reviewer 2 Report

-